# Dynamics Changes of Microorganisms Community and Fermentation Quality in Soybean Meal Prepared with Lactic Acid Bacteria and *Artemisia argyi* through Fermentation and Aerobic Exposure Processes

**DOI:** 10.3390/foods11060795

**Published:** 2022-03-10

**Authors:** Weiwei Wang, Zhongfang Tan, Lingbiao Gu, Hao Ma, Zhenyu Wang, Lei Wang, Guofang Wu, Guangyong Qin, Yanping Wang, Huili Pang

**Affiliations:** 1School of Physics and Microelectronics, Zhengzhou University, Zhengzhou 450052, China; bingzhi213608@163.com (W.W.); mahaoworks@foxmail.com (H.M.); 2Henan Key Lab Ion Beam Bioengineering, School of Agricultural Sciences, Zhengzhou University, Zhengzhou 450052, China; tzhongfang@zzu.edu.cn (Z.T.); 18438699518@163.com (Z.W.); qinguangyong@zzu.edu.cn (G.Q.); wyp@zzu.edu.cn (Y.W.); 3School of Biological and Food Engineering, Anyang Institute of Technology, Anyang 455000, China; gulingbiao@foxmail.com; 4Academy of Animal Science and Veterinary Medicine, Qinghai University, Xining 810016, China; wanglei382369@163.com (L.W.); jim963252@163.com (G.W.)

**Keywords:** fermented soybean meal, additives, fermentation characteristics, microbial communities, mycotoxins

## Abstract

This study evaluated the effects of *Lactiplantibacillus plantarum* subsp. *plantarum* ZA3, *Artemisia argyi* and their combination, on the fermentation characteristics, microbial community, mycotoxins and crude flavonoids content of fermented soybean meal during fermentation (under anaerobic conditions) and aerobic exposure (under aerobic conditions). The results showed that ZA3, *Artemisia argyi* and ZA3+ *Artemisia argyi* groups had lower pH values and higher lactic acid concentrations compared with controls, and additives increased the abundance of *Lactiplantibacillus* and decreased those of *Acetobacter* and *Enterobacter*; in particular, *Artemisia argyi* and ZA3+ *Artemisia argyi* reduced the abundance of fungi, such as *Aspergillus*, *Pichia*, *Fusarium*, *Cladosporium* and *Xeromyces*. Meanwhile, the contents of mycotoxins were lower in treated groups, and even mycotoxins in the control were significantly reduced after 30 d (*p* < 0.05). Crude flavonoids that were correlated positively with *Lactococcus* and negatively with *Bacillus*, *Aspergillus*, *Enterobacter* and *Kazachstania* were significantly higher in the *Artemisia argyi* and ZA3+ *Artemisia argyi* groups (*p* < 0.05).

## 1. Introduction

In response to the difficulties and challenges faced by the livestock industry, fermented feed has shown great promise; one important reason for this is that fermented feed is made from a wide range of raw materials that can be turned into valuable products through fermentation, avoiding competition for food between humans and animals and solving the problem of resource shortage. Soybean meal (SBM) is a kind of biomass resource left after soybean oil is extracted, and it has the advantages of a crude protein (CP) content of 30–50%, reasonable amino acid composition, high digestibility and good palatability [1]. However, the utility of conventional SBM is limited by macromolecular proteins, pathogenic microorganisms, mycotoxins and many antinutritional factors; therefore, achieving full usage of SBM would be meaningful both for the efficient development of animal husbandry and for effective use of waste resources. 

To date, the main studies on SBM have focused on the degradation of macromolecular proteins and antinutritional factors [2], and the most widely used methods to resolve the above problems include physical, chemical and solid-state fermentation (SSF). Compared to the other two treatments, SSF is widely used because of its environmental protection, large-scale production and high degradation rate [3]. However, a number of issues regarding SBM fermentation, such as the microbial flora that are responsible for the quality of fermentation during fermentation, were only reported by Chen et al. [4] and Wang et al. [5], not to mention what happens after fermentation, and the effect of fermentation on toxins. 

Due to cost and other factors, SBM raw materials are usually not sterilized in production, and this creates a high potential for contamination with refuse during the fermentation process. It is, therefore, essential to use exotic additives to facilitate the fermentation process and thus, maximize the fermentation quality. Lactic acid bacteria (LAB), which are considered to have potential benefits for the host and are widely used as additives for feed fermentation, have also been studied as an inoculant in the fermentation of SBM. Additionally, LAB including *Enterococcus* (*E.*) *faecium*, *E. faecalis*, *Lactiplantibacillus* (*L.*) *brevis*, *L. casei*, *L*. *plantarum* and *L. buchneri* have been shown to improve the quality of fermented soybean meal (FSBM) through the promotion of lactic acid production and reduction of pH [6], increased contents of isoflavone, total peptide and free amino nitrogen [7] and reduced antinutritional factors [8]. Shi et al. [9] found SSF with *E. faecium* could improve the quality of SBM by increasing small peptides and free amino acid; Yang et al. [10] improved the protein quality and degraded allergen in SBM by *L. casei* fermentation and enzymatic hydrolysis; *L. buchneri* could increase acetic acid content and aerobic stability of fermented feeds containing SBM [11]; Heng et al. [12] found that added *L*. *plantarum* and *L. rhamnosus* to ferment SBM could increase the contents of small peptides and organic acid, and lower the pH value. 

*Artemisia argyi* (AA), a perennial (sometimes semi-shrubby) plant that contains a variety of active compounds, such as essential oils, polysaccharides and flavonoids, and has been shown to have antifungal activity and strong antibacterial effects against *Staphylococcus aureus*, *Escherichia coli* and *Salmonella enteritidis* [13], is a traditional Chinese medicine herb that is widely distributed in Asia and used to treat diseases including diarrhea, hemostasis and inflammation [14]. In recent years, AA has been actively applied as a feed additive in animal production, as antibiotics used in feed have caused animal intestinal microbiota dysbiosis, drug residues and environmental pollution and have been banned in many countries, including China, which has completely banned the use of all growth additives except Chinese herbal medicine in feed since July 2020 [15]. Additionally, AA has a unique aroma that can fend off mosquitoes, flies and mites to improve the breeding environment and increase the palatability and feed intake of animal diets [16]. Moreover, most of current AA is wild and distributed in wilderness areas and making reasonable use of AA resources as a feed additive can improve the quality of the feed, lower feed cost and reduce environmental pollution. 

At present, the typical fermentation time of SBM is 3 d, leading to problems, such as insufficient acid production and high pH value, meaning that presumably harmful bacteria are not sufficiently inhibited. Application of AA in feed is mostly in the form of green or dried powder for direct feeding. Thus, with the goal of revealing the fermentation mechanism of SBM treated with LAB and AA, making the most of SBM and AA resources and exploring efficient feed additives, this study explored the effects of adding LAB and AA to SBM on the fermentation quality, chemical composition, microbial flora and mycotoxins of FSBM during both 3 days (d) and 30 d fermentation and its corresponding post-opening aerobic exposure phases.

## 2. Materials and Methods

### 2.1. Materials and Fermentation Preparation

SBM was purchased from COFCO Corporation (Shijiazhuang, China), whole plant of dry *Artemisia argyi* (AA) was supplied by Anyang Institute of Technology and crushed for use. *L. plantarum* subsp. *plantarum* ZA3 was isolated from healthy weaned piglets fecal matter with broad-spectrum activity against a wide range of microorganisms and kept in our lab [17], and ZA3 was added at a rate of 12% at 1 × 10^8^ colony forming units (cfu)/mL and AA was 8% (weight/weight, *w*/*w*). Experimental treatments were designed as follows: (1) CK (control check), SBM: water = 1:1; (2) ZA3, ZA3 + CK; (3) AA, AA + CK, and (4) AA + ZA3, AA + ZA3 + CK. After SBM was mixed homogenously with additives, 500 g of each treatment was packed manually into plastic film bags (Dragon N-6, Asahi Kasei Co., Tokyo, Japan), vacuumed, and sealed with a vacuum sealer (P-290, Shineye, Dongguan, China). A total of 224 bags including 10 anaerobic sampling stages (12 h, 24 h, 36 h, 48 h, 60 h, 3 d, 7 d, 12 d, 18 d and 30 d) and four aerobic exposures (3 d–3 d, 3 d–7 d, 30 d–3 d and 30 d–7 d), with each treatment performed in quadruplicate for each sampling period, were prepared and kept at room temperature, which the minimum and maximum through the whole experiment periods from 20 August 2020 to 25 September 2020 in Zhengzhou, Henan province was 16 and 36 °C. Moreover, “3 d–3 d” and “3 d–7 d” means SBM that fermented under anaerobic for 3 d, was opened and exposed to air for 3 and 7 d, while “30 d–3 d” and “30 d–7 d” was anaerobic for 30 d and then exposed to air for 3 and 7 d, respectively.

### 2.2. pH, Chemical Composition, Crude Flavonoids and Microbial Population Analysis of Soybean Meal and Artemisia argyi

pH value was measured using a pH meter (Mettler Toledo Co., Ltd., Greifensee, Switzerland). According to standard procedures detailed by the Association of Official Analytical Chemists (AOAC) [18], dry matter (DM) was determined by an oven, CP by the Kjeldahl method using DigiPREP TKN Systems (UDK159, VELP, Milan, Italy), ether extract (EE) through Soxhlet extractor method by SER148 EE analyzer (SER148, VELP, Milan, Italy), and crude fiber (CF) was based on Van Soest et al. [19] by using a FIWE3/6 CF analyzer (FIWE3/6, VELP, Milan, Italy).

Crude flavonoid content was determined using the slightly modified procedure of spectrometry provided by Hudz et al. [20] and made calibration curves for rutin. The crude flavonoids were extracted using ethanol aqueous solution assisted with ultrasonics, and the ratio of sample and 95% ethanol was 1:20 (g:mL), extraction time was 60 min and ultrasonic power was 500 W. After the extract was centrifuged at 6000 rpm for 10 min, 3.0 mL of supernatant added to 0.3 mL 20% NaNO_2_ solution left for 3 min, then 0.4 mL 40% Al (NO_3_)_3_ solution was added, mixed and left for another 3 min, next 1.0 mL 20% NaOH solution was added and mixed, and the absorbance of the reaction mixtures was measured at 510 nm by a UV-Visible Spectrophotometer (UV mini-1240, SHIMADZU, Tokyo, Japan) after standing for 10 min.

The microbiological population analysis referred to the method of Pang et al. [21]. For each sample, 10 g was taken and diluted with 90 mL distilled water, then the supernatant was diluted serially to 10-fold and inoculated in triplicate on different agar plates: Man Rogosa Sharpe (MRS) agar to enumerate LAB, Potato Dextrose Agar (PDA) containing 0.15% tartaric acid for yeast and mold, Eosin Methylene Blue (EMB) agar for coliform bacteria, Nutrient Agar (NA) for aerobic bacteria and bacillus, and Clostridium enrichment medium (CLO) agar for clostridium, respectively. Among them, Bacillus and Clostridium were spread on NA and CLO medium after incubation at 75 °C in a water bath for 15 min, moreover, except MRS which was under anaerobic condition, all others were under aerobic conditions. After incubating at 37 °C for 48 h (LAB, coliform bacteria, aerobic bacteria, Bacillus and Clostridium) and 60 h (yeast and mold), colonies were counted as the numbers of viable microorganisms in cfu/g of fresh matter (FM).

### 2.3. Evaluation of Sensory and Surface Structure of Soybean Meal and Fermented Soybean Meal 

The color, smell and texture of SBM and FSBM during fermentation for 3 and 30 d, and aerobic exposure of 7 d were evaluated according to Meinlschmidt et al. [22]. After the FSBM samples were unsealed, sensory profiling was performed using descriptive sensory analysis: the samples were weighed into dishes, and the sensory evaluation team was composed of five people to evaluate the colors, odor and hand feel characteristics by eye observation, nose smell and hand touch. Physical properties and microstructure changes of SBM and FSBM during fermentation for 3 and 30 d were examined by scanning electron microscopy (SEM) (Helios G4 CX, Thermo Scientific, Waltham, MA, USA) at ×2000-fold magnification, respectively.

### 2.4. Fermentation Quality, Chemical Composition and Microbial Population Analysis of Fermented Soybean Meal

At each opening, three bags were randomly selected and opened, 10 g samples were suspended with 90 mL distilled water and then filtered to determine organic acids by using high-performance liquid chromatography (HPLC) (Waters Alliance e2695, Waters, Milford, MA, USA). 

pH value, chemical composition, and microbial population analyses of FSBM were the same as the description in Section 2.2.

### 2.5. Bacterial and Fungal Community Analysis

SBM and FSBM fermented for 3 and 30 d and aerobic exposure of 7 d were sampled to analyze the bacterial and fungal communities. Total genome DNA from samples was extracted using CTAB/SDS method, and DNA concentration and purity were monitored on 1% agarose gels. The bacterial 16S rDNA (16S V4) was amplified using primers 515F (5′-GTGCCAGCMGCCGCGGTAA-3′) and reverse primer 806R (5′-GGACTACHVGGG TWTCTAAT-3′), fungi ITS was amplified using the ITS1F (5′-CTTGGTCATTTAGAGGAAGTAA-3′) and ITS2-2043R (5′-GCTGCGTTCTTCATC GATGC-3′). All PCR reactions were carried out with Phusion^®^ High-Fidelity PCR Master Mix (New England Biolabs Ltd., Beijing, China), and PCR products were detected by 2% agarose gel electrophoresis. Samples with the bright main strip between 400–450 bp were chosen and purified with Qiagen Gel Extraction Kit (Qiagen, Dusseldorf, Germany). Sequencing libraries were generated using TruSeq^®^ DNA PCR Sample Preparation Kit (Illumina, San Diego, CA, USA) and index codes were added, libraries quality was assessed on the Qubit@ 2.0 Fluorometer (Thermo Fisher Scientific Inc., Carlsbad, CA, USA) and Agilent Bioanalyzer 2100 system (Agilent Technologies, Santa Clara, CA, USA), and sequenced at Shanghai Applied Protein Technology Co., Ltd. (Shanghai, China). on IlluminaHiSeq2500 platform and 250 bp paired-end reads were generated. 

Sequence analyses were performed by Uparse software (Uparse v7.0.1001, http://drive5.com/uparse/ accessed on 24 May 2021), and sequences with 97% similarity were assigned to the same operational taxonomic units (OTUs). Alpha diversity was applied in analyzing the complexity of species diversity for a sample through Chao and Shannon using QIIME (Version 1.7.0) and displayed with R software (Version 2.15.3, http://www.mothur.org/wiki/Chao and http://www.mothur.org/wiki/Shannon accessed on 12 July 2021). 

PCoA analysis was demonstrated for the variance of bacterial and fungi communities’ structure by weighted gene co-expression network analysis (WGCNA), stat and ggplot2 packages in R software (version 2.15.3), and spearman correlation heatmaps based on the Spearman correlation coefficients among bacterial and fungi communities and fermentation parameters were produced. The data were analyzed using the free online Lingbo Microclass (http://www.biomicroclass.com accessed on 5 November 2021).

### 2.6. Mycotoxin and Crude Flavonoids Analysis

Aflatoxin B1 (AFB1), deoxynivalenol (DON), zearalenone (ZEN), ochratoxin (OA) and fumonisin (FUM) were analyzed using enzyme linked immunosorbent assay (ELISA) kits provided by Lianshuo Biological Technology Co., Ltd. (AMEKO, Shanghai, China). 

Crude flavonoids analysis were using the method described in Section 2.2.

### 2.7. Statistical Analysis

The data of fermentation characteristics, microbial population and chemical composition were subjected to analysis using the IBM SPSS statistical package 22.0 (SPSS Inc., Chicago, IL, USA), and Student–Newman–Keuls multiple range tests were used to evaluate differences among treatments and the significance was declared at *p* < 0.05. 

## 3. Results and Discussion

### 3.1. Characteristics of Soybean Meal and Artemisia argyi 

Generally, the fermentation process and final feed quality are largely influenced by the characteristics of the material. As shown in Table 1, the pH value of the SBM was 6.7, while AA had a relatively lower pH value of 5.6. The DM concentration for SBM and AA were 882.1 and 371.6 g/kg, while the CP, CF and EE contents were 43.8, 6.8 and 1.6 (%DM) for SBM, and for AA were 10.9, 53.8 and 3.1 (%DM), respectively. The content of crude flavonoids in AA was 23.7 mg g/kg DM and was 0.6 in SBM, a mere fraction of AA. This is also consistent with reports that more than 100 chemical components have been isolated from AA, including flavonoids, polysaccharides, terpenoids, volatile oils and aromatic aldehydes, of which flavonoids are the main group of active ingredients [23].

The epiphytic microorganisms on the surface of a raw material determine the quality of the fermentation, and for high-quality fermented feed, the number of LAB should reach 5.0 lg cfu/g FM of raw material. In the present study, the populations of LAB on SBM and AA were 7.0 and 5.6 lg cfu/g FM, but higher amounts of harmful bacteria, such as aerobic bacteria, coliform bacteria and bacilli were also observed in both of the two raw materials at more than 4.5 lg cfu/g FM, which would result in poor fermentation without inoculants. Therefore, it was necessary to add additives to inhibit these harmful microorganisms in order to ensure the fermentation quality.

### 3.2. Sensory Evaluation and Surface Structure of Fermented Soybean Meal

As shown in Figure 1a, after 3 d and 30 d of fermentation, the ZA3 group always maintained the light yellowish brown color of the raw material, but with a fluffier texture and a stronger sour aroma compared with the other groups; the colors of the AA and ZA3 + AA groups were sandy beige, and these groups had a fluffy texture and a rich aroma of AA itself. After 3 d of fermentation followed by 7 d of aerobic exposure, both CK and AA groups condensed into a mass and CK produced a strong foul smell; only CK was agglomerated after 30 d of fermentation and then 7 d of aerobic exposure, for which the reason may be that the environment quickly changed to aerobic, which made aerobic yeast and mold proliferate rapidly until they became the dominant flora and finally caused FSBM spoilage.

From the sensory aspect alone, it is clear that treatment groups showed improved quality of FSBM, and this observation was further verified by SEM. Figure 1b shows the surface images of SBM and FSBM at a magnification factor of ×2000, from which it can be seen that SBM had relatively large, compact and smooth-faced structures; the FSBM after 3 d of fermentation in the CK group showed a flaky structure; and in the treated groups had small cracked structures and holes. After 30 d of fermentation, large fragment structures were detected only in CK, and treated groups had more loose networks with diffuse and big holes, which was more pronounced after 30 d of fermentation. This is also consistent with the reports of Chen et al. [4] and Wang et al. [5], who observed that the surface structure of FSBM showed fragmental, cracked, small, structures and large holes compared with uninoculated SBM by SEM. The changes in surface structure may be due to the fact that lignocellulose was decomposed by extracellular enzymes (carbohydrate, protease and amylase), and FSBM was easier to digest than SBM, maybe because soybean proteins with diffuse structures and big holes have higher solubility [24]. 

### 3.3. Fermentation Quality, Chemical Composition and Microbial Population of Fermented Soybean Meal

#### 3.3.1. Fermentation Quality

Figure 2 displays the significant effect of the treatment × day interactions on pH and organic acids (*p* < 0.001). At 24 h of fermentation, the pH in ZA3 and ZA3 + AA had decreased to 4.75, below that in SBM, and for AA it was also significantly reduced at 36 h (*p* < 0.05) (Figure 2a) and dropped to 4.43 after 3 d. Combined with the information described in Table 1, the pH of the AA raw sample was already as low as 5.6, and fermentation further lowed the pH and created an acidic environment at the early stages of fermentation by direct acidification. After 3 d of fermentation followed by 7 d of aerobic exposure, the pH in CK raised again to 7.0, which was even higher than the SBM value of 6.7 and the highest among all of the groups by a significant amount (*p* < 0.05), while the additive groups were all below 4.4 (Figure 2b). These results may be because, after the fermented feed was opened and came into contact with the air, carbohydrates, organic acids, proteins and amino acids were decomposed by aerobic microorganisms, heat was generated and the pH increased [25]; LAB and AA, on the other hand, exerted an antibacterial effect, slowing down the proliferation of aerobic bacteria and thus, prevented the rapid increase in pH. In contrast to SBM and 3 d of fermentation FSBM, pH values in all groups decreased significantly after 30 d of fermentation; moreover, treated groups had lower pH than CK (*p* < 0.05). The above results show that 30 d of fermentation can better create a low-pH environment that is not conducive to the growth of miscellaneous bacteria. 

For organic acids, during fermentation and aerobic exposure stages, butyric acid was detected only in the CK group. The highest values for both lactic acid and acetic acid were found in the ZA3 group with 120.8 and 15.4 g/kg DM after 30 d of fermentation; as for the aerobic exposure phase, the AA and ZA3 + AA groups showed significantly increased contents of lactic acid and acetic acid (*p* < 0.05) (Figure 2c–f), which all corresponded to the above observation that pH decreased significantly in these groups during corresponding periods. Furthermore, the CK group showed a significant reduction in lactic acid content during the exposure period after 30 d of fermentation (*p* < 0.05) which may account for the rapid rise in pH at this stage. Acetic acid, which can improve the aerobic stability of fermented feed through significant antifungal properties [26], had consistently higher levels in all the treated groups than that in the CK after 30 d of fermentation and subsequent aerobic exposure, which may be one of the reasons for the better aerobic stability of the treatment groups. Guan et al. [27] also found that heterofermentative *L. buchneri* and *L. rhamnosus* improved fermented feed aerobic stability by producing higher levels of acetic acid. In short, both LAB and AA had positive effects on fermentation properties in terms of lowering pH value and increasing lactic acid accumulation over the whole fermentation and aerobic exposure stages, especially at 30 d. 

#### 3.3.2. Chemical Composition 

As shown in Figure 3, CP, CF and EE were significantly affected by days, treatments and day × treatment interactions (*p* < 0.001). The change of CP is one of the most important parameters in FSBM, which is often hydrolyzed into peptides, amino acids, and ammonia by proteases and microbial activity. In the present study, the content of CP in FSBM at 30 d was greater in all groups than that at 3 d, which may be mainly due to the fact that during the fermentation process, beneficial bacteria consume part of the organic material by respiration, thus releasing CO_2_ and H_2_O and reducing the total amount of products, resulting in a “concentration effect” of the protein. The protein levels in ZA3 were significantly higher both after the first 7 d (Figure 3a) and after 30 d of fermentation and then 3 d of aerobic exposure (*p* < 0.05) (Figure 3b), and similar effects were also reported by Su et al. [28] that after SSF, the contents of CP and trichloroacetic-acid-soluble protein increased (*p* < 0.05) by 14.28% and 25.53%, respectively. This is probably a result of the bacteriological proteins and inorganic ammonium salts of the fermentation process being converted by LAB and the plant protein in SBM being metabolized and utilized by microorganisms to be converted into bacterial proteins, which also changed the quality of the proteins in the SBM even more. 

AA and ZA3 + AA treatments significantly affected the CF and EE contents during both fermentation and aerobic exposure (*p* < 0.05). In particular, a black oily substance that was presumed to be the volatile oil in AA was observed on the measuring cups of SER148 EE analyzer during EE testing in the AA and ZA3 + AA groups, while CK and ZA3 were transparent in contrast. As some of the multiple pharmacological activities AA possesses are attributed to this volatile oil, the presence of this substance and CF would stimulate mastication, gastrointestinal motility, enrich the gastrointestinal tract and regulate the microbiota of the gastrointestinal tract [29].

#### 3.3.3. Microbial Population 

The dynamics of the microbial populations in FSBM listed in Figure 4 all showed a trend of increase followed by a decrease. After 36 h of fermentation, the LAB counts were higher in groups ZA3 (10.5 lg cfu/g FM) and ZA3 + AA (10.6 lg cfu/g FM) than those in CK and AA (*p* < 0.05); correspondingly, aerobic and coliform bacteria were significantly reduced in the former two groups (*p* < 0.05), which were in line with the aforementioned rapid decreases in pH values in this stage. Inoculants of LAB and AA, as previously mentioned, can produce a faster accumulation of lactic acid and lower pH values, inhibiting the growth of undesirable bacteria. After 3 d of fermentation, bacilli and aerobic and coliform bacteria in all additive groups were significantly lower than that in the CK (*p* < 0.05); moreover, these three types of bacteria plus yeast and mold were significantly increased in CK during all subsequent aerobic exposure stages (*p* < 0.05), which directly resulted in an increase in ambient temperature and pH value. The fatty acids produced by the yeast may have inhibited the growth of LAB, eventually leading to FSBM deterioration. Compared to 3 d of fermentation, all counts of microorganisms were significantly lower at the end of 30 d, especially in the ZA3 group, due to the restriction of rapid acidification and antagonistic activity. Additionally, clostridium was only detected in the CK group after 30 d of fermentation followed by 7 d of aerobic exposure, and the fermentation products of clostridium are mainly butyric acid, which can produce unpleasant odors, high calories and energy loss and reduced quality of fermented feed [30]. This is also consistent with the detection of butyric acid in the CK group only. Several studies have proven that AA has a strong inhibitory effect on bacteria and fungi including *Aspergillus flavus*, *Escherichia coli* and *Colletotrichum fragariae* [13,31]. Thus, the ZA3 + AA treatment used in the present study can inhibit undesirable bacteria and thereby reduce excessive spoilage of FSBM at the aerobic exposure stage.

### 3.4. Microbial Communities, Diversities and Relative Abundances in Fermented Soybean Meal

#### 3.4.1. Microbial Communities and Diversities

The diversity, function and structure of microbial communities are important microbial ecology topics and have been the focus of considerable research efforts. Figure 5 shows the communities and diversities of bacteria and fungi in SBM and FSBM, and microbial abundances and diversities in a single sample were reflected by the Alpha diversity. After 3 d and 30 d of fermentation, chao1 indexes showed that the richness of bacteria and fungi had no significant differences within the groups, while ZA3 and ZA3 + AA showed significantly reduced chao1 indexes of bacteria, and AA, a significantly reduced index of fungi from 7 d of aerobic exposure until the end of the experiment (*p* < 0.05). Even when exposed for 7 d after 30 d of fermentation, the chao1 indexes of both bacteria in ZA3 and fungi in AA remained decreased.

Diversities shown by Shannon indexes indicated that both bacterial and fungal communities decreased significantly in the ZA3 + AA group after the whole fermentation and aerobic exposure stages (*p* < 0.05), and bacteria in ZA3 and fungi in AA also presented the same results. The antibacterial properties of ZA3 need not be described, considering activities of acidification and antagonism towards other bacteria by AA, which promotes a reduction in fungal diversity, and is in agreement with previous reports that AA has a strong inhibitory effect on fungi [32].

FSBM with LAB and AA inoculants showed decreased richness and diversity of bacterial and fungal communities, which may be because LAB rapidly proliferated to become the dominant species in the bacterial communities and reduced counts of aerobic bacteria in fermented feed; in other words, the complex microbial communities in SBM and AA were gradually replaced by LAB after fermentation, which is a requirement for successful fermentation in general [33]. Moreover, although the AA group did not show a significant decline in fungi through the plate method, chao1 and Shannon indexes showing the richness and diversities of fungi were significantly reduced after 30 d of fermentation, which also illustrates the importance of combining culture methods with high-throughput methods. 

#### 3.4.2. Relative Abundances at the Phylum Level

Exploring the dynamic changes of species and numbers of dominant microorganisms in fermentation processes is helpful in understanding the fermentation process and improving the quality of fermentation [34], and, based on what is known from the survey, this is the first report concerning the bacterial and fungal changes in FSBM. Figure 6 shows the bacterial and fungal relative abundances in SBM and FSBM. At the phylum level, for bacteria (Figure 6a), the dominant microorganism of SBM was *Cyanobacteria* (93.61%), a Gram-negative bacterium, of which five types have been identified as toxin producers. *Proteobacteria*, a group of Gram-negative commensal bacteria, reside in the intestine as *Firmicutes*, influencing metabolism and maintaining the homeostasis of the internal environment through their specific flora structure, activity and metabolites [35]. After 3 d of fermentation, the numbers of *Firmicutes* and *Proteobacteria* increased and dominated the fermentation among all the groups, with only relative abundance showing differences between them. *Firmicutes* in CK was 37.43%, and in the treated groups was above 86.78%, reaching 94.88% in group ZA3, and it was significantly higher in all the treated groups than in CK (91.81%) after aerobic exposure for 7 d. After 30 d of fermentation, *Firmicutes* in CK increased to 70.26%, while no significant changes were observed compared to 3 d in the treatment groups. This is because the pH values in all the groups decreased significantly after 30 d of fermentation, and the acidic and anaerobic environments could produce acids and a variety of enzymes, which was conducive to the growth of Gram-positive *Firmicutes*. *Firmicutes* has many beneficial members, such as *L. plantarum* and are one of the most common bacterial groups among the gut microbiota [36]. Furthermore, there are also reports that *Firmicutes* and *Proteobacteria* can degrade cellulose in anaerobic environments, and the presence of these two phyla may be an important cause of variation in the DM and organic fractions of the FSBM [37]. The bacterial community during the fermentation of SBM showed a dynamic change from Gram-negative to Gram-positive, indicating the presence of abundant pathogenic bacteria in the raw material, which indicates LAB was beneficial to feed fermentation, replacing the harmful bacteria that were abundant in raw materials as the dominant bacteria after fermentation.

The structural change and quantitative growth of fungal communities are closely related to fermentation quality and mycotoxins [38]. In the present study, in SBM, the dominant fungi at the phylum level were *Ascomycota* (66.18%) and *Basidiomycota* (31.35%) (Figure 6c). *Ascomycota* is the most diverse and species-rich phylum in the kingdom of fungi, comprising approximately 11,000 species. It shows a broad range of life modes, such as pathogenic (agriculturally and clinically), saprobic and endophytic [39]. After 3 d of fermentation, *Ascomycota* was reduced to 58.92 in CK while values in the treated groups were all below 50%, moreover, its abundances in treated groups were all below 35% after aerobic exposure of 7 d; after 30 d of fermentation, *Ascomycota* was significantly reduced in all groups (*p* < 0.05), dropping to 49.71% in CK and even as low as 32.55 and 30.44% in ZA3 and AA. This is also consistent with the absence of fungi from 7 to 30 d of fermentation according to the culture method. Direct feeding of anaerobic fungi including *Basidiomycota* can increase feed intake, weight gain and milk production in ruminants [40]. In this study, at the end of the 3 d of fermentation, *Basidiomycota* was 14.26% in CK, while in treated groups it was at least 26.70%; after aerobic exposure for 7 d, it reduced to 0.03% in CK, but in treated groups remained above 25%, with these values lasting to the end of the 30 d of fermentation. Furthermore, AA had a significant effect on the increase of *Basidiomycota* in FSBM (*p* < 0.05), making it very suitable for use as a fermented feed additive.

#### 3.4.3. Relative Abundances at the Genus Level

At the early fermentation stage, cocci that produce lactic acid are usually the dominant LAB, which can start lactic acid fermentation. At the genus level, for bacteria (Figure 6b), treated groups showed increased abundance of *Lactiplantibacillus* and decreased abundance of *Enterobacter* compared with the levels in CK, and AA had a higher relative abundance of *Weissella* (17.41%), *Lactococcus* (15.75%) and *Pediococcus* (6.49%); moreover, from 3 d of fermentation to 7 d of aerobic exposure, the highest abundance of *Lactiplantibacillus* (95.65%) was found in ZA3 + AA, while *Enterobacter* increased from 16.87% to 28.58% in CK and *Weissella* increased to 48.73% in AA, respectively. Lactobacilli grow rapidly and dominate as fermentation proceeds and the pH decreases. After 30 d of fermentation, *Weissella* decreased to 4.38% and *Enterobacter* increased to 38.86% in CK, while *Lactiplantibacillus* exhibited a relative abundance of 57.49% in AA, reaching the highest level among the four groups at 84.84% after 7 d of aerobic exposure. In summary, the majority of LAB species in fermentation belonged to *Lactiplantibacillus*, *Weissella*, *Lactococcus* and *Pediococcus* in this research. These are desirable functional bacteria during fermentation and have been widely used to improve fermentation quality by producing lactic acid, reducing pH and inhibiting the growth of undesirable bacteria [30]. *Enterobacter* is generally considered to be undesirable during fermentation because these bacteria can ferment lactic acid to acetic acid, succinic acid and some endotoxins, which can cause degradation of fermentation quality and feed contamination [41]. In addition, *Acetobacter*, which can change the ethanol into acetic acid and reduce the fermentation quality of fermented feed [30], was 9.93% in CK after 7 d of aerobic exposure.

For fungi (Figure 6d), *Aspergillus*, *Fusarium* and *Cladosporium* were dominant in SBM. About 20 species of the genus *Aspergillus* are known to be harmful to humans and other animals, the most infamous being *Aspergillus flavus*, which produces aflatoxin and can be inhibited by AA [42]. CK had the highest content of *Aspergillus* and the AA group had the least during the fermentation and aerobic exposure stages. Numerous species of *Fusarium* are important plant pathogens that can produce ranges of secondary metabolites and cause opportunistic mycoses [43]. *Aspergillus* and *Fusarium* were higher in CK than in treated groups with 9.46% and 9.98%, respectively. *Pichia*, *Aspergillus*, *Fusarium*, *Cladosporium*, *Xeromyces* and *Acremonium* increased significantly in CK and ZA3 compared with AA and ZA3 + AA following 7 d of aerobic exposure after 3 and 30 d of fermentation (*p* < 0.05). The main consequence of *Cladosporium* species in foods is spoilage and discoloration [44]; *Xeromyces* can spoil foods with little water availability and *Acremonium* produces the weak trichothecene crotocin and the metabolite cerulenin, which enhances aflatoxin biosynthesis [45]. The relative abundance of *Kazachstania* was higher in CK compared to the treated groups, and the AA group contained the least *Kazachstania* and *Pichia* among all the groups after 30 d of fermentation in this research. *Kazachstania* and *Pichia*, which grow rapidly and increase the pH of the environment when exposed to air and accelerate the growth of undesirable microorganisms, are the most commonly detected yeast in aerobic spoiled fermented feed [46]; moreover, they might play important roles in aerobic deterioration of feed [47]. These also explain the aforementioned fact that in the AA group, no mold was detected, and communities and diversities were significantly reduced. In addition, at 30 d, *Aspergillus*, which maintained a relatively low abundance in AA and ZA3 + AA groups, was reduced significantly in CK, with a relative abundance of 1.27% compared with 3 d. This may be an important factor in the reduction of mycotoxin content after 30 d of fermentation (Figure 8). All the above findings again suggest that AA is suitable as a feed additive to inhibit fungi, especially during aerobic exposure.

### 3.5. Correlation Analyses of the Bacterial and Fungal Communities with Fermentation Properties of Fermented Soybean Meal 

Figure 7 illustrates the relationships among the top 10 most abundant bacterial and fungal genera and fermentation properties. As can be seen in Figure 7a,e, after 3 d of fermentation, pH was correlated positively with *Aspergillus* (*r* = 0.60; *p* < 0.05) and negatively with *Flavobacterium*, *Cladosporium*, *Acremonium* and *Alternaria* (*p* < 0.01); meanwhile, lactic acid was correlated positively with *Lactobacillus* (*r* = 0.89; *p* < 0.001) and *Flavobacterium* (*r* = 0.63; *p* < 0.05) and negatively with *Enterobacter* (*r* = −0.94; *p* < 0.001), *Weissella*, *Pediococcus* and *Lactococcus* (*p* < 0.01), and negative correlations between *Pediococcus*, *Weissella*, *Lactococcus* and lactic acid concentration were observed. As a homofermentative LAB, inoculant *L. plantarum* subsp. *plantarum* ZA3 would dominate at the outset of fermentation and be outcompeted in low-pH conditions by cocci, which convert 6-carbon sugars containing six carbon atoms into two 3-carbon atoms of lactic acid. Lactic acid leads to a rapid reduction in pH, and the acidic environment can inhibit other harmful microorganisms on the raw materials and preserve the feed as well as possible [48]. In the present study, after aerobic exposure for 7 d (Figure 7b,f), pH was positively correlated with the genera *Enterobacter* (*r* = 0.61; *p* < 0.05) and *Aspergillus* (*r* = 0.66; *p* < 0.05), and for organic acid, lactic acid was positively correlated with *Kazachstania* (*r* = 0.71; *p* < 0.01); this may be because *Kazachstania* can assimilate lactic acid and hydrolyze glucuronide as a metabolic substrate for heterofermentative LAB, facilitating the use of fructose for acetic acid production, and *Kazachstania* dominates the yeast genus in low-pH environments. Acetic acid was correlated positively with *Flavobacterium* and *Pichia* (*p* < 0.01) and negatively with *Lactococcus* (*r* = −0.67; *p* < 0.05). *Lactiplantibacillus* was correlated positively with lactic acid and negatively with pH value; meanwhile, after 30 d of fermentation (Figure 7c,g), pH was correlated positively with *Xeromyces* (*r* = 0.75; *p* < 0.01) and *Kazachstania* (*r* = 0.69; *p* < 0.05) and negatively with *Paenibacillus* and *Acremonium* (*p* < 0.05); lactic acid was correlated positively with *Fusarium* and *Xeromyces* (*p* < 0.05) and negatively with *Flavobacterium* (*r* = −0.68; *p* < 0.05). After aerobic exposure for 7 d (Figure 7d,h), pH was correlated positively with *Pichia* (*r* = 0.60; *p* < 0.05), *Aspergillus* (*r* = 0.72; *p* < 0.01) and *Xeromyces* (*r* = 0.85; *p* < 0.001) and negatively with *Flavobacterium* (*r* = −0.79; *p* < 0.01) and *Paenibacillus* (*r* = −0.80; *p* < 0.01); for organic acid, lactic acid was correlated positively with *Lactiplantibacillus* (*r* = 0.76; *p* < 0.01), and acetic acid was correlated positively with *Flavobacterium* (*r* = 0.74; *p* < 0.01) and *Paenibacillus* (*r* = 0.77; *p* < 0.01) and negatively with *Fusarium* (*r* = −0.64; *p* < 0.05), *Xeromyces* (*r* = −0.73; *p* < 0.01) and *Acremonium* (*r* = −0.58; *p* < 0.05). The succession of fungi in FSBM after exposure to air is generally started by yeast with a pH increase, thereby accelerating the growth of some acid nonresistant undesirable microorganisms. Acetic acid was correlated positively with *Pichia* following aerobic exposure for 7 d after 3 d of fermentation, possibly because *Pichia* is one of the acid-assimilating yeast that consumes organic acid to increase the pH [49], but it was correlated negatively with *Fusarium*, *Xeromyces* and *Acremonium* following aerobic exposure for 7 d after 30 d of fermentation, as acetic acid generally inhibits the growth of undesirable bacteria and thereby prevents aerobic spoilage of feed.

### 3.6. Toxin Content 

Mycotoxins are secondary metabolites produced by different fungal species, and contamination with mycotoxins during feed storage is widespread [50]. It is known that feed containing mycotoxins will reduce animal performance, including decreased feed intake, increased incidence of disease and impairment of the immune system [51]. The government of China has set maximum contents (MCs) for the most common mycotoxins in feed, including AFB1, DON, ZEA, OA and FUM. 

The contents of toxins after fermentation and aerobic exposure are shown in Figure 8. After 3 d of fermentation, AFB1, DON, ZEN, OA and FUM all had the highest contents in CK compared with treated groups with 10.39 μg/kg, 1.16 mg/kg, 1.24 mg/kg, 120.08 μg/kg and 27.28 mg/kg (*p* < 0.05), respectively; although these levels were greater than the maximum specified added contents, they were still much lower than those in SBM with 14.63 μg/kg, 1.62 mg/kg, 1.46 mg/kg, 163.17 μg/kg and 33.78 mg/kg. After 30 d of fermentation, contents of all five mycotoxins in the CK were significantly reduced compared with 3 d (*p* < 0.05), while the AFB1, DON, OA and FUM contents in ZA3 were the lowest with 1.52 μg/kg, 0.50 mg/kg, 19.11 μg/kg and 0.44 mg/kg DM, respectively. AFB1 is the most stable and toxic of mycotoxins contaminating feed under natural conditions. Research has shown that some LAB are specific for the adsorption of food mutagens, including AFB1 [52], and that the adsorption capacity varies according to the strain and dose of the LAB [53]. The degradation of AFB1 by ZA3 in this study, relative to CK, reached 95.22%. In the present research, after 30 d of fermentation, the ZA3 + AA group had the lowest ZEN level of 0.002 mg/kg DM compared to all other groups, even ZA3 or AA alone, presumably due to the combined effect of both strain and AA. Additionally, after 7 d of continued aerobic exposure, although AFB1, DON, ZEN, OA and FUM levels in CK rose again to 11.61 μg/kg, 0.58 mg/kg, 0.45 mg/kg, 37.31 μg/kg and 1.08 mg/kg DM, which also significantly decreased compared to that at 7 d of aerobic exposure after 3 d of fermentation 23.59 μg/kg, 2.25 mg/kg, 0.73 mg/kg, 194.21 μg/kg and 25.87 mg/kg DM (*p* < 0.05), respectively. This is a good indication that 30 d of fermentation was beneficial in reducing the contents of toxins in FSBM compared to 3 d, even under natural fermentation conditions.

### 3.7. Determination of Crude Flavonoids and Linkages with Microbial Communities

Flavonoids act as a resistance barrier produced by hosts against phytopathogens. Natural products rich in flavonoids have been documented to have good antibacterial effects and antibiotic-resistant bacteria remain sensitive to these compounds. In livestock production, the addition of flavonoids to livestock feed can significantly increase resistance to disease and improve the immunity of the animal [54]. Crude flavonoids evaluated during fermentation and aerobic exposure are shown in Figure 9a, and the contents in AA and ZA3 + AA were significantly higher than those in CK and ZA3 both after 3 d and 30 d of fermentation (*p* < 0.05). Previous reports have found that crude flavonoids have effective antibacterial activities against fungi, such as *Fusarium*, *Aspergillus*, *Penicillium* and some yeasts [55,56], which also explains why mold and yeast populations were significantly suppressed during fermentation and aerobic exposure stages in AA and ZA3 + AA groups. 

Linkages between microbial abundances and crude flavonoids are displayed in Figure 9b–e. After both 3 d of fermentation and the subsequent 7 d of aerobic exposure, crude flavonoids were correlated positively with *Lactococcus* and negatively with *Aspergillus*; meanwhile, at 3 d and 30 d, both were positively correlated with *Acremonium* and *Alternaria*, two genera which were negatively correlated with pH in the description above, and negatively with *Xeromyces*, one of the genera positively correlated with pH. *Flavobacterium*, which is one of the dominant flora in the animal intestine, was correlated positively with organic acids and negatively with pH, and was negatively and positively correlated, respectively, with flavonoids following aerobic exposure for 7 d after 3 d and 30 d of fermentation. This also showed that crude flavonoids from AA can inhibit undesirable microorganisms to improve the quality of fermented feed during the fermentation and aerobic exposure process, and confirmed that 30 d of fermentation could produce further improvement in the quality of FSBM compared to 3 d.

## 4. Conclusions

This study evaluated the fermentation quality, chemical composition, microorganism community and mycotoxins of short (12, 24, 36, 48, 60 h and 3 d), long fermentation (7, 12, 18 and 30 d), and their corresponding aerobic exposure stages after openings for FSBM. The results revealed compared to 3 d fermentation, *Lactiplantibacillus plantarum* subsp. *plantarum* ZA3 and *Artemisia argyi* treated group at 30 d fermentation and subsequent exposure had more loose networks with diffuse and big holes in appearance, lower pH values and undesirable microorganism abundance, and higher organic acids contents, especially significant to reduce mycotoxins. To sum up, 30 d fermentation fermented soybean meal treated with ZA3 and *Artemisia argyi* is optimal. Even considering the cost reasons, in the actual production with large quantities, 30 d of fermentation can be used as a means of preserving FSBM. The results of this research provide a preliminary reference towards establishing the possibility that lactic acid bacteria in complex with *Artemisia argyi* might be used as efficient feed additives.

## Figures and Tables

**Figure 1 foods-11-00795-f001:**
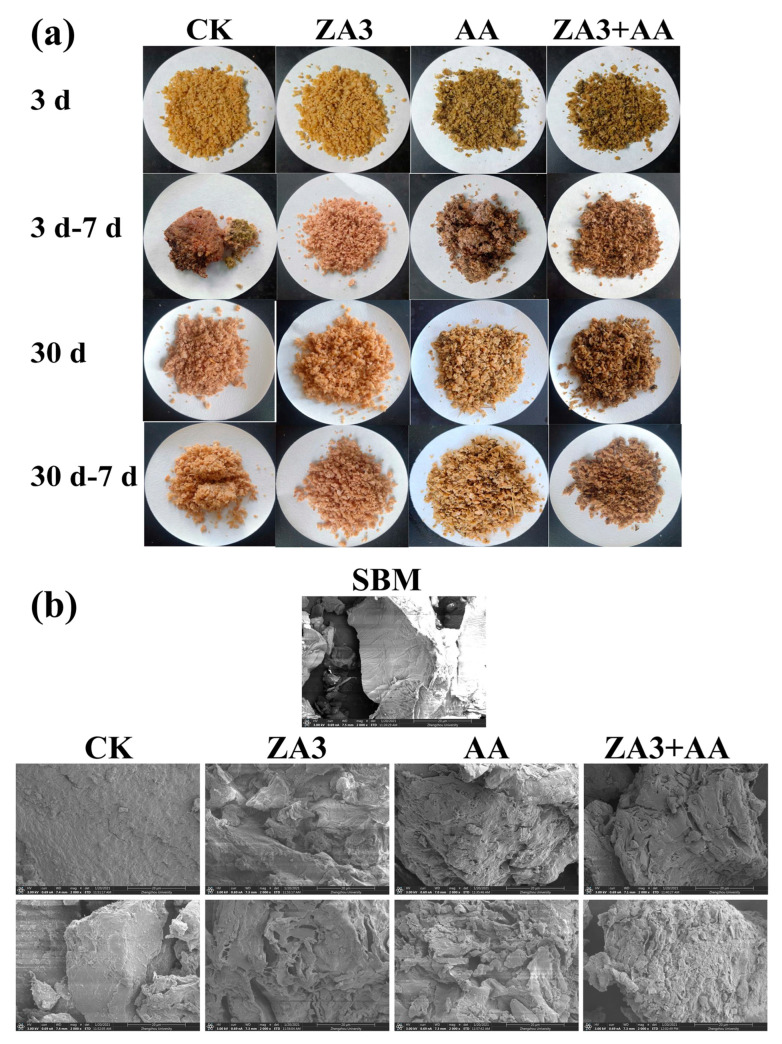
Sensory evaluation and surface structure of FSBM during fermentation and aerobic exposure. (**a**) The appearance of FSBM after 3 and 30 d fermentation and 7 d exposure after these two stages, (**b**) SEM images after 3 d and 30 d at ×2000-fold magnification. d, days.

**Figure 2 foods-11-00795-f002:**
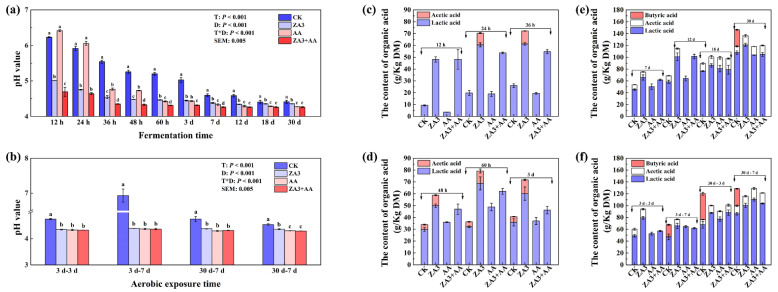
Fermentation quality of FSBM during fermentation and aerobic exposure. (**a**,**b**), pH value; (**c**–**f**), organic acid. Different lowercase letters (a–d) for the same treatment time indicate significant differences (*p* < 0.05) among different treatment groups according to Student–Newman–Keuls test. The significant difference between days and treatment interactions is expressed as *p* < 0.001. DM, dry matter; h, hours; d, days.

**Figure 3 foods-11-00795-f003:**
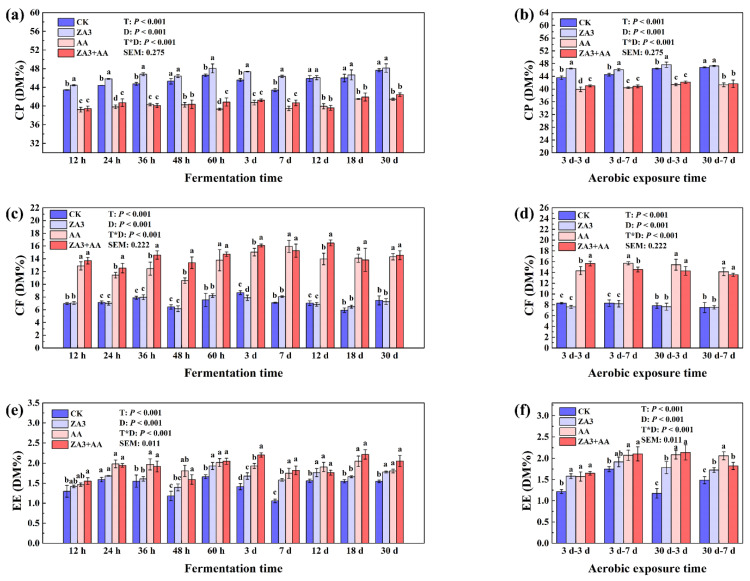
Chemical composition of FSBM during fermentation and aerobic exposure. (**a**,**b**), CP, crude protein; (**c**,**d**), CF, crude fiber; (**e**,**f**), EE, ether extract; DM, dry matter. Different lowercase letters (a–d) for the same treatment time indicate significant differences (*p* < 0.05) among different treatment groups according to the Student–Newman–Keuls test. The significant difference between days and treatment interactions is expressed as *p* < 0.001. h, hours; d, days.

**Figure 4 foods-11-00795-f004:**
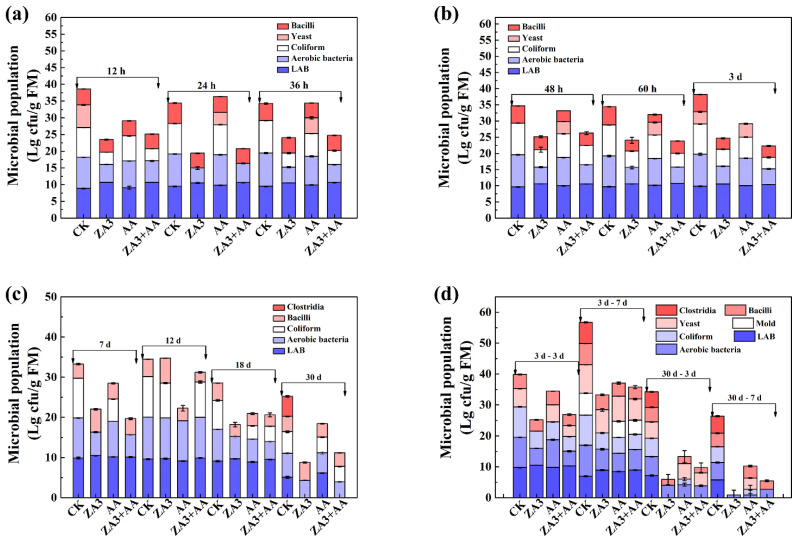
Microbial population of FSBM during fermentation and aerobic exposure. (**a**), fermented for 12, 24 and 36 h; (**b**) fermented for 48, 60 and 72 h; (**c**) fermented for 7, 12, 18 and 30 d; (**d**) aerobic exposure for 3 d–3 d, 3 d–7 d, 30 d–3 d and 30 d–7 d. cfu, colony forming unit; FM, fresh material; h, hours; d, days.

**Figure 5 foods-11-00795-f005:**
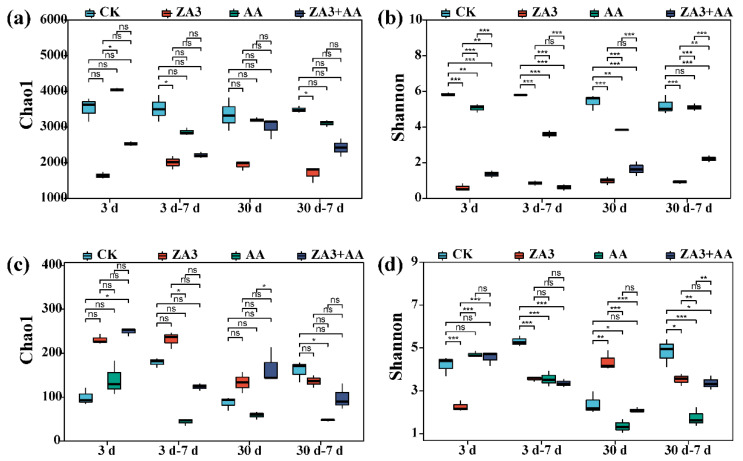
Alpha diversity of bacteria and fungi of FSBM during fermentation and aerobic exposure. (**a**,**b**), Chao1 and Shannon indices of bacterial communities; (**c**,**d**), Chao1 and Shannon indices of fungi communities. ns, no significance. *, *p* < 0.05; **, *p* < 0.01; ***, *p* < 0.001. d, days.

**Figure 6 foods-11-00795-f006:**
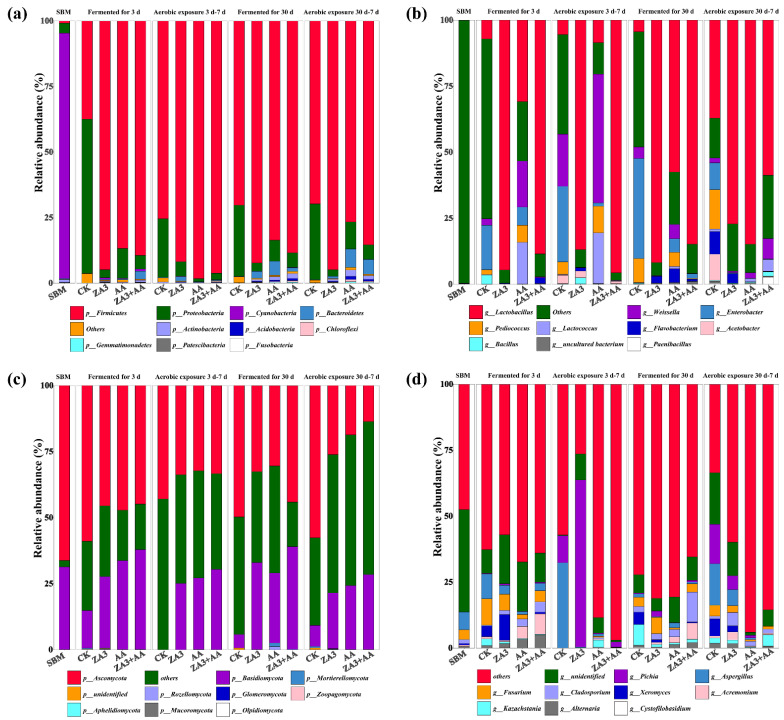
The bacterial and fungal communities of FSBM during fermentation and aerobic exposure. The phylum level of bacterial and fungal communities is shown in (**a**,**c**); and the genus level is shown in (**b**,**d**). d, days.

**Figure 7 foods-11-00795-f007:**
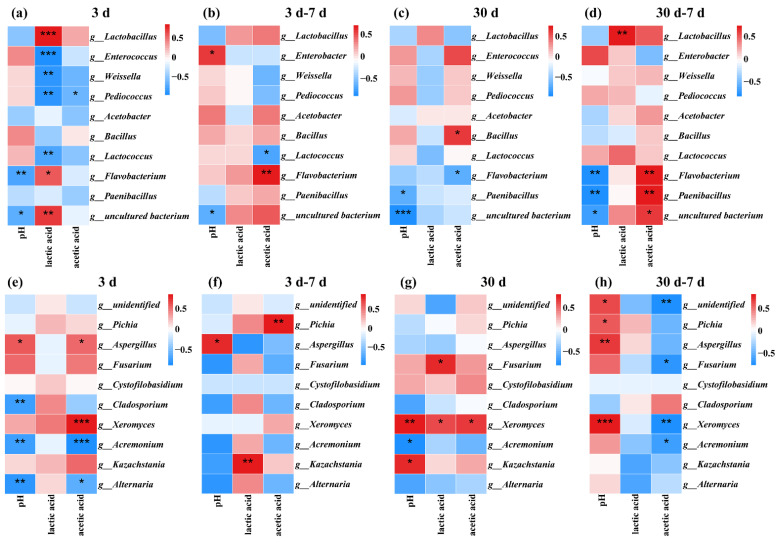
Spearman correlation heatmap of abundance of the top 10 enriched bacteria and fungi at the genus level with fermentation properties during fermentation and aerobic exposure. Bacterial: 3 d (**a**) and 30 d (**c**) of fermentation, 3 d–7 d (**b**) and 30 d–7 d (**d**) of aerobic exposure; Fungi, 3 d (**e**) and 30 d (**g**) of fermentation, and 3 d–7 d (**f**) and 30 d–7 d (**h**) of aerobic exposure. Positive correlations are shown in red, and negative correlations are shown in blue. *, *p* < 0.05; **, *p* < 0.01; ***, *p* < 0.001. d, days.

**Figure 8 foods-11-00795-f008:**
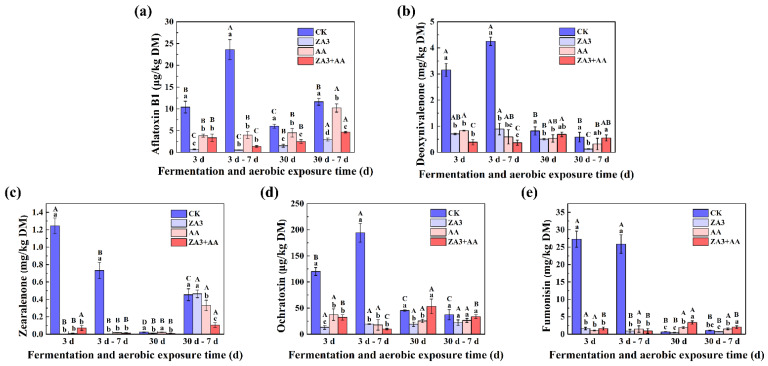
Content of toxin during fermentation and aerobic exposure. (**a**), aflatoxin B1 (μg/kg DM); (**b**), deoxynivalenone (mg/kg DM); (**c**), zearalenone (mg/kg DM); (**d**), fumonisin (mg/kg DM) and (**e**), ochratoxin (μg/kg DM). Different lowercase letters (a–d) for the same treatment time indicate significant differences (*p* < 0.05) among different treatment groups. Different capital letters (A–D) for the same treatment groups indicate significant differences among different treatment times (*p* < 0.05) according to Student–Newman–Keuls test. d, days.

**Figure 9 foods-11-00795-f009:**
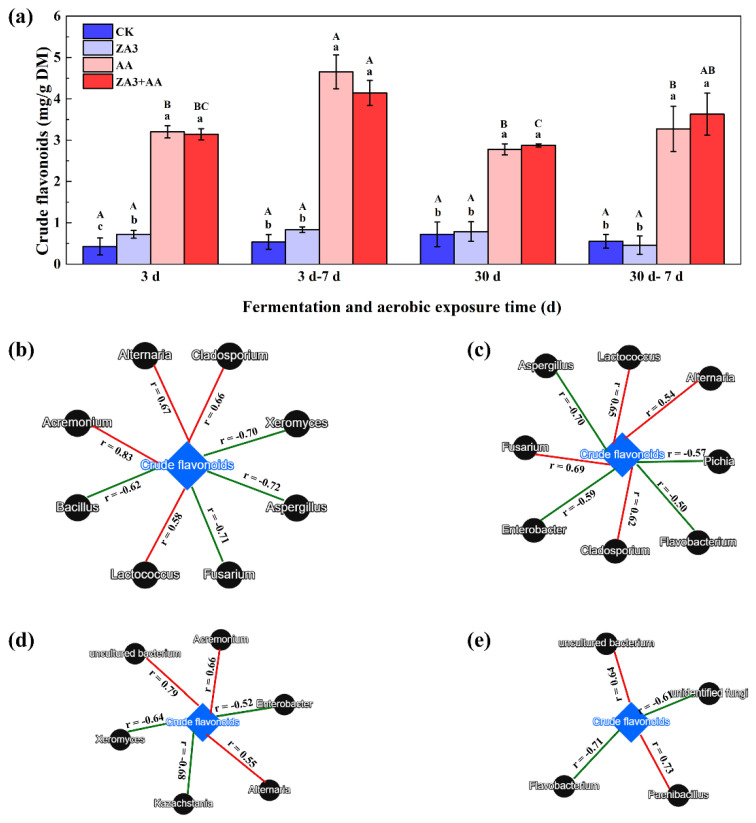
Content of crude flavonoids and linkages with microbial during fermentation and aerobic exposure. (**a**), crude flavonoids (mg/g DM). (**b**–**e**), co-occurrence networks analysis the top 10 enriched bacteria and fungi at the genus level with crude flavonoids in 3 d and 30 d of fermentation, and 3 d–7 d and 30 d–7 d of aerobic exposure. Different lowercase letters (a–d) for the same treatment time indicate significant differences (*p* < 0.05) among different treatment groups. Different capital letters (A–D) for the same treatment groups indicate significant differences among different treatment times (*p* < 0.05) according to Student–Newman–Keuls test. d, days. Each co-occurrence pair among top 10 enriched bacteria and fungi at the genus level and crude flavonoids has an absolute Pearson rank correlation above 0.50 [red straight line, positive correlation (r ≥ 0.50); green straight line, negative correlation (r ≤ −0.50)]. Microbes are shown by black circle shaped nodes, and crude flavonoids are shown by blue rhombus nodes.

**Table 1 foods-11-00795-t001:** Chemical composition and microbial population of SBM and AA (Mean ± SD, *n* = 3).

Item	SBM	AA
pH	6.7 ± 0.02	5.6 ± 0.01
DM (g/kg)	882.1 ± 2.81	371.6 ± 3.77
CP (%DM)	43.8 ± 0.02	10.9 ± 0.02
CF (%DM)	6.8 ± 0.07	53.8 ± 0.84
EE (%DM)	1.6 ± 0.07	3.1 ± 0.08
Crude flavonoids (mg/g DM)	0.6 ± 0.06	23.7 ± 1.84
LAB (lg cfu/g FM)	7.0 ± 0.02	5.6 ± 0.08
Aerobic bacteria (lg cfu/g FM)	8.4 ± 0.23	5.8 ± 0.05
Coliform bacteria (lg cfu/g FM)	5.0 ± 0.13	5.6 ± 0.46
Bacilli (lg cfu/g FM)	4.5 ± 0.19	6.6 ± 0.01

SD, standard deviation; DM, dry matter; CP, crude protein; CF, crude fiber; EE, ether extract; LAB, lactic acid bacteria; cfu, colony forming unit; FM, fresh material.

## Data Availability

The data generated from the study is clearly presented and discussed in the manuscript.

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
