# Peer review of "Dynamics Changes of Microorganisms Community and Fermentation Quality in Soybean Meal Prepared with Lactic Acid Bacteria and Artemisia argyi through Fermentation and Aerobic Exposure Processes"

_foods, 2022, doi:10.3390/foods11060795_

Round 1
Reviewer 1 Report
The title should be improved.
The document presents several grammatical and contextual errors. The methodology is not clear, which does not make it easy to read.
Abstract. The name of Lactobacillus plantarum, it is suggested to change to the new nomenclature.
The objective is confused, the fermentation is a condition? aerobic conditions is other treatment? or the fermentation was under anerobic condition?. If it is correct, I suggest to improve the wording of the text.
In the objective, is necessary added the change on the microorganism community.
Define “CK”
L90. Delete inoculation
L93: What is m/m?
Room temperature of 16-36 °C, is very wide range, this drastically modifies the growth of microorganisms, why was the temperature not controlled?
L99-101. the times and conditions are confused, Could the authors improve the explanation of their fermentation conditions? What are the 14 times?
Section 2.4, What is fermentation characteristics?
The methodology is confused, Could the authors outline the methodology of the work carried out?
What selective mediums were used?
The color, How was evaluated? Using L, a, b ?
I not understand, what is 30-d-3-d, 30-d-7-d?
What would be the problem to be solved with this product?
In general, what would be the proposal? under what conditions can it be proposed that FSBM can be fermented?
14 times × 4 treatments × 4 replicates? How is it?
Author Response
Reviewer 1
Comments and Suggestions for Authors
Point 1: The title should be improved.
Thanks for the good suggestion. We have changed the title into “Dynamics changes of microorganisms community and fermentation quality in soybean meal prepared with lactic acid bacteria and Artemisia argyi through fermentation and aerobic exposure processes”, please see P1, L2-5.
Point 2: The document presents several grammatical and contextual errors. The methodology is not clear, which does not make it easy to read.
We are very sorry. We have carefully revised the grammar and structure of manuscript, and performed a professional language revision, the following figure is the proof. We also have re-written the methodology section, please see P2-4, L96-165.
Point 3: Abstract. The name of Lactobacillus plantarum, it is suggested to change to the new nomenclature.
Thank you very much. We have revised all “Lactobacillus (L.) plantarum subsp. plantarum” into “Lactiplantibacillus (L.) plantarum subsp. plantarum”, please see Lactiplantibacillus in whole revised manuscript.
Point 4: The objective is confused, the fermentation is a condition? aerobic conditions is other treatment? or the fermentation was under anerobic condition? If it is correct, I suggest to improve the wording of the text.
We are very sorry for the confusing description we have provided. In this study, “fermentation” and “aerobic condition” are two different treatment conditions, “fermentation” was under anerobic condition, and “aerobic exposure” was the exposure of sealed bags to aerobic conditions when opened after 3 and 30 days of fermentation, respectively. We have explained it and revised the objective, please see P1, L21-22.
Point 5: In the objective, is necessary added the change on the microorganism community.
Thanks for the good suggestion. We have added the change on the microorganism community in the objective, please see P1, L20.
Point 6: Define “CK”
We are very sorry that we did not clarify the term “CK” in the manuscript, which may also bring some readers reading doubts. “CK” means “Control check”, which is a commonly used expression for control group in some research articles, such as the following references. “CK” in this research means “soybean meal without additives”, and we have added the explanation of CK in the manuscript, please see P3, L104.
Zhao, J.; Dong, Z.; Li, J.; Chen, L.; Bai, Y.; Jia, Y.; Shao, T. Ensiling as pretreatment of rice straw: The effect of hemicellulase and Lactobacillus plantarum on hemicellulose degradation and cellulose conversion. Bioresource Technol. 2018, 266, 158-165. doi:10.1016/j.biortech.2018.06.058.
Yin, Y.; Pereira, J.; Zhou, L.; Lorenzo, J.M.; Tian, X.; Zhang, W. Insight into the effects of sous vide on cathepsin b and l activities, protein degradation and the ultrastructure of beef. Foods. 2020, 9, 1441. doi: 10.3390/foods9101441.
Zhao, S.; Yang, F.; Wang, Y.; Fan, X.; Feng, C.; Wang, Y. Dynamics of fermentation parameters and bacterial community in high-moisture alfalfa silage with or without lactic acid bacteria. Microorganisms. 2021, 9, 1225. doi:10.3390/microorganisms9061225.
Point 7: Delete inoculation
Thank you, we have deleted inoculation in revised manuscript.
Point 8: What is m/m?
We are so sorry that due to our negligence there were something went wrong, and thank you for reminding us. “m/m” means “mass/mass”, and we intended to express the weight ratio of Artemisia argyi and soybean meal. We have changed “m/m” into “weight/weight, w/w”, please see P3, L103.
Point 9: Room temperature of 16-36 °C, is very wide range, this drastically modifies the growth of microorganisms, why was the temperature not controlled?
Thank you, and 16-36 °C was the minimum and maximum room temperature through the whole fermentation and aerobic exposure periods from 20th of August, 2020 to 25th of September, 2020 in Zhengzhou, Henan province. Room temperature was chosen because the objective of this study was to investigate the effect of adding lactic acid bacteria and Artemisia argyi on fermented soybean meal under natural fermentation conditions, which would satisfy the needs of practical production better and thus didn't control the temperature. We have explained in manuscript. Please see P3, L111-113.
Point 10: I not understand, what is 30 d-3 d, 30 d-7 d ?
We are very sorry that we did not provide much information about “30 d-3 d” and “30 d-7 d” in the manuscript. “30 d-3 d” and “30 d-7 d” means soybean meal was exposed to aerobic for 3 d and 7 d after fermentation of 30 d, for example, “30 d-3 d” means soybean meal that fermented under anaerobic for 30 d, was opened and exposed to air for 3 d, while “3 d-3 d” was soybean meal fermented under anaerobic for 3 d and then exposed to air for 3 d. We have added detailed instructions about “3 d-3 d”, “3 d-7 d”, “30 d-3 d” and “30 d-7 d” in manuscript, please see P3, L113-116.
Point 11: The times and conditions are confused, Could the authors improve the explanation of their fermentation conditions? What are the 14 times?
We are sorry for providing the confusing description. In the experimental design, “14 times” means that 14 sampling sessions, including 10 anaerobic fermentation stages (12 h, 24 h, 36 h , 48 h, 60 h, 3 d, 7 d, 12 d, 18 d and 30 d) and 4 aerobic fermentation stages (3 d-3 d, 3 d-7 d, 30 d-3 d and 30 d-7 d). We have added the explanation in the appropriate place of manuscript, please see P3, L108-110.
Point 12: 14 times × 4 treatments × 4 replicates? How is it?
We are sorry. For experimental design, “14 times” means 14 sampling sessions including 10 anaerobic fermentation stages and 4 aerobic fermentation stages as reply in point 11, “4 treatments” means the experiment was divided into 4 groups: (1) CK (control check), SBM: water=1:1; (2) ZA3, ZA3+CK; (3) AA, AA+CK, and (4) AA+ZA3, AA+ZA3+CK, and “4 replicates” means each treatment was performed in quadruplicate for each sampling period. Thus, totally 224 (14 times × 4 treatments × 4 replicates = 224) bags were made, and 3 bags were randomly selected from each treatment group and opened for analysis of various indicators at each sampling.
Although this expression follows the probabilistic approach to research, making each sampling more random and representative, and has appeared in some fermentation related literature, such as the following study: a total of 24 bags (2 groups×4 treatments×3 replicates) were prepared and kept at ambient temperature (20–35 °C) (Ni, K.; Wang, X.; Lu, Y.; Guo, L.; Li, X.; Yang, F. Exploring the silage quality of alfalfa ensiled with the residues of astragalus and hawthorn. Bioresour Technol. 2020, 297, 122249, doi:10.1016/j.biortech.2019.122249.), a total of 72 silos was used for this study divided into 4 treatments and 3 replicates for 6 fermentation days (4*3*6 = 72) (Ali, N.; Wang, S.; Zhao, J.; Dong, Z.; Li, J.; Nazar, M.; Shao, T. Microbial diversity and fermentation profile of red clover silage inoculated with reconstituted indigenous and exogenous epiphytic microbiota. Bioresour Technol. 2020, 314, 123606, doi:10.1016/j.biortech.2020.123606.) and 36 bags (3 treatments × 4 days × 3 repeats) were obtained and maintained at an ambient temperature (26-30 °C) (Lv, H.; Pian, R.; Xing, Y.; Zhou, W.; Yang, F.; Chen, X.; Zhang, Q. Effects of citric acid on fermentation characteristics and bacterial diversity of Amomum villosum silage. Bioresour Technol. 2020, 307, 123290, doi:10.1016/j.biortech.2020.123290.), this may lead to misunderstandings for readers, so we deleted it in the revised manuscript.
Point 13: Section 2.4, What is fermentation characteristics?
We are so sorry that due to our negligence there were something went wrong, thank you for reminding us, and we have changed “fermentation characteristics” into “fermentation quality”, please see P4, L158.
Point 14: The methodology is confused, Could the authors outline the methodology of the work carried out?
Thanks for the good question, we have modified the method, please see “materials and methods” from P2, L96 to P5 L205.
Point 15: What selective mediums were used?
Yes, we have added selective mediums in the revised manuscript, please see P3, L136-147. For each sample, 10 g was taken and diluted with 90 mL distilled water, then the super-natant was diluted serially to 10-fold and inoculated in triplicate on different agar plates: Man Rogosa Sharpe (MRS) agar to enumerate LAB, Potato Dextrose Agar (PDA) containing 0.15% tartaric acid for yeast and mold, Eosin Methylene Blue (EMB) agar for coliform bacteria, Nutrient Agar (NA) for aerobic bacteria and bacillus, and Clostridium enrichment medium (CLO) agar for clostridium, respectively. Among which, bacillus and clostridium was spreaded on NA and CLO medium after incubation at 75 °C water bath for 15 min, moreover, except MRS was under anaerobic condition, others were all under aerobic. After incubated at 37 °C for 48 h (LAB, coliform bacteria, aerobic bacteria, bacillus and clostridium) and 60 h (yeast and mold), colonies were counted as the numbers of viable microorganisms in cfu/g of fresh matter (FM).
Point 16: The color, How was evaluated? Using L, a, b ?
Thank you for the reminder, we have added the corresponding information to the manuscript, please see P3-4, L149-154.
The color, smell and texture of SBM and FSBM during fermented for 3 and 30 d, and aerobic exposure of 7 d were evaluated according to Meinlschmidt et al. [22]. After the FSBM samples were unsealed, sensory profiling was performed using descriptive sensory analysis: the samples were weighed into dishes, and the sensory evaluation team was composed of five people to evaluate the colors, odor and hand feel characteristics by eye observation, nose smell and hand touch.
Meinlschmidt, P.; Ueberham, E.; Lehmann, J.; Schweiggert-Weisz, U.; Eisner, P. Immunoreactivity, sensory and physico-chemical properties of fermented soy protein isolate. Food Chem. 2016, 205, 229-238. doi: 10.1016/j.foodchem.2016.03.016.
Point 17: What would be the problem to be solved with this product?
Thank you very much. Researches on fermented soybean meal (FSBM) now is focused on the degradation of macromolecular proteins and anti-nutritional factors, however, for fermented feed, microbial composition and toxin content are also very important. This study found FSBM with Lactiplantibacillus plantarum subsp. plantarum ZA3 and Artemisia argyi could reduce the abundance of undesirable microorganisms and the content of mycotoxins, increase lactic acid bacteria and organic acid, and finally improve fermentation quality. Especially compared to conventional 3 d fermentation, these indicators are more significant at 30 d. Moreover, after exposed to oxygen for 3 or 7 d subsequent to 3 and 30 d of fermentation, ZA3+AA groups still maintained lower pH value, undesirable microorganisms and mycotoxins contents, higher organic acid concentration and lactic acid bacteria abundance and crude flavone content. Therefore, the products prepared in this study can solve the problem of high mycotoxins and pathogenic microorganisms, low organic acids and beneficial microorganisms contents in SBM; in addition, metabolic substances beneficial to the health of livestock and poultry produced in the fermentation process, will improve the palatability and nutritional value of soybean meal, partially or completely replace the amount of antibiotics, and 30 d fermentation can be used as a means of storing FSBM, which can provide some references for relevant scholars who study FSBM.
Point 18: In general, what would be the proposal? under what conditions can it be proposed that FSBM can be fermented?
Thank you. This study evaluated the fermentation quality, chemical composition, microorganism community and mycotoxins of short (12, 24, 36, 48, 60 h and 3 d), long fermentation (7, 12, 18 and 30 d), and their corresponding aerobic exposure stages after openings for FSBM. The results revealed compared to 3 d fermentation, Lactiplantibacillus plantarum subsp. plantarum ZA3 and Artemisia argyi treated group at 30 d fermentation and subsequent exposure had more loose networks with diffuse and big holes in appearance, lower pH values and undesirable microorganisms abundance, and higher organic acids contents, especially significant to reduce mycotoxins. To sum up, 30 d fermentation fermented soybean meal treated with ZA3 and Artemisia argyi is optimal. Even considering the cost reasons, in the actual production with large quantities, 30 d of fermentation can be used as a means of preserving FSBM. The results of this research provide a preliminary reference towards establishing the possibility that lactic acid bacteria in complex with Artemisia argyi might be used as efficient feed additives.
We have added this proposal into manuscript, please see P18-19, L628-640.

Reviewer 2 Report
Manuscript Number: foods-1615354
Title: Dynamics changes of microorganisms community and fermentation quality in fermentation and aerobic exposure processes of soybean meal prepared with lactic acid bacteria and Artemisia argyi
Overview and general recommendation
The article structure is compact, sequential and logical. The data are adequate to support the conclusion. The methods section provides sufficient information on sampling, definitions, data collection and data analysis. References are up-dated adequate and correctly cited.
In this study the author evaluated the effects of Lactobacillus plantarum, a plant Artemisia argyi and their combination on fermentation characteristics, mycotoxins and crude flavonoid content of fermented soybean meal during fermentation and aerobic exposure and they recommend their combined use as efficient feed additives.
Minor comments:
- The part of Abstract: In this part I recommend not to use abbreviations.
- What does CK abbreviation?
- The part of Introduction: In my opinion, this part should be improved with several bibliographic references that present the current state of research in the field of study of authors. For example:
- Line 50: about “have rarely been reported”: references are missing
- Line 56: “also has been studied” – just 2 references! etc.
- The part of Results and Discussion:
- In almost all subchapters there are no results from the literature compared to the results of research conducted by authors: 3.2, 3.3.1, 3.3.2, 3.3.3 etc.
- I recommend do not be use abbreviation in subtitles. For example:
- Line 172 – “Characteristics of SBM and AA”
- Line 193 – “Sensory evaluation and surface structure of FSBM” etc.
- All figures should be enlarged so that the data in them can be seen.
- The part of Conclusion:
- Just like the abstract there should be no more abbreviations in this part.
The authors carry out an interesting work and I recommend it minor revision.
Author Response
Reviewer 2
Comments and Suggestions for Authors
Point 1: The part of Abstract: In this part I recommend not to use abbreviations
Thanks for the good suggestion. We have revised the abstract following the suggestion. Please see P1, L19-31.
Point 2: What does CK abbreviation?
We are very sorry that we did not clarify the term CK in the manuscript, which may also bring some readers reading doubts. CK means “Control check”, which is a commonly used expression for control group in some research articles, such as the following references. CK in this research means “soybean meal without additives”, and we have added the explanation of CK in the manuscript, please see P2, L104.
Zhao, J.; Dong, Z.; Li, J.; Chen, L.; Bai, Y.; Jia, Y.; Shao, T. Ensiling as pretreatment of rice straw: The effect of hemicellulase and Lactobacillus plantarum on hemicellulose degradation and cellulose conversion. Bioresource Technol. 2018, 266, 158-165. doi:10.1016/j.biortech.2018.06.058.
Yin, Y.; Pereira, J.; Zhou, L.; Lorenzo, J.M.; Tian, X.; Zhang, W. Insight into the effects of sous vide on cathepsin b and l activities, protein degradation and the ultrastructure of beef. Foods. 2020, 9, 1441. doi: 10.3390/foods9101441.
Zhao, S.; Yang, F.; Wang, Y.; Fan, X.; Feng, C.; Wang, Y. Dynamics of fermentation parameters and bacterial community in high-moisture alfalfa silage with or without lactic acid bacteria. Microorganisms. 2021, 9, 1225. doi:10.3390/microorganisms9061225.
Point 3: The part of Introduction: In my opinion, this part should be improved with several bibliographic references that present the current state of research in the field of study of authors. For example:
Line 50: about “have rarely been reported”: references are missing
Line 56: “also has been studied” – just 2 references! etc.
Thank you very much. We have revised and added relevant information in respective places. Please see P2, L50-54, L65-71.
However, a number of issues regarding SBM fermentation, such as microbial flora that are responsible for the quality of fermentation during fermentation, only be reported by Chen et al. [4] and Wang et al. [5], not to mention what happens when it be opened after fermentation, and the effect of fermentation on toxins.
- Chen, L.; Zhao, Z.; Yu, W.; Zheng, L.; Li, L.; Gu, W.; Xu, H.; Wei, B.; Yan, X. Nutritional quality improvement of soybean meal by Bacillus velezensis and Lactobacillus plantarum during two-stage solid- state fermentation. AMB Express. 2021, 11, 23, doi:10.1186/s13568-021-01184-x.
- Wang, C.; Shi, C.; Su, W.; Jin, M.; Xu, B.; Hao, L.; Zhang, Y.; Lu, Z.; Wang, F.; Wang, Y., et al. Dynamics of the physico-chemical characteristics, microbiota, and metabolic functions of soybean meal and corn mixed substrates during two-stage solid-state fermentation. mSystems 2020, 5, doi:10.1128/mSystems.00501-19.
Shi et al. [9] found SSF with E. faecium could improve the quality of SBM by increasing small peptides and free amino acid; Yang et al. [10] improved the protein quality and degrade allergen in SBM by L. casei fermentation and enzymatic hydrolysis; L. buchneri could increase acetic acid content and aerobic stability of fermented feeds containing SBM [11]; Heng et al. [12] found that added L. plantarum and L. rhamnosus to ferment SBM could increase the contents of small peptides and organic acid, lower the pH value.
- Shi, C. ; Zhang, Y. ; Yin, Y. ; Wang, C. ; Wang, Y. Amino acid and phosphorus digestibility of fermented corn-soybean meal mixed feed with Bacillus subtilis and Enterococcus faecium fed to pigs. J. Anim. Sci. 2017, 95, 3996-4004. doi: 10.2527/jas2017.1516.
- Yang, H.; Qu, Y.; Li, J.; Liu, X.; Wu, R.; Wu, J. Improvement of the protein quality and degradation of allergens in soybean meal by combination fermentation and enzymatic hydrolysis. LWT-Food Sci. Technol. 2020, 128, 109442, doi: 10.1016/j.lwt.2020.109442.
- Nishino, N.; Hattori, H. Resistance to aerobic deterioration of total mixed ration silage inoculated with and without homofermentative or heterofermentative lactic acid bacteria. J. Sci. Food Agr. 2007, 87, 2420-2426. doi:10.1002/jsfa.2911.
- Heng, X.; Chen, H.; Lu, C.; Feng, T.; Li, K.; Gao, E. Study on synergistic fermentation of bean dregs and soybean meal by multiple strains and proteases. LWT-Food Sci. Technol. 2022, 154, 112626, doi:10.1016/j.lwt.2021.112626.
Point 4: In almost all subchapters there are no results from the literature compared to the results of research conducted by authors: 3.2, 3.3.1, 3.3.2, 3.3.3 etc.
Thanks for the good suggestion. We have added the related information, please see 3.2, 3.3.1, 3.3.2, 3.3.3 etc.
This is also consistent with the reports of Chen et al. [4] and Wang et al. [5], who observed that the surface structure of FSBM showed fragmental, cracked, small, cracked structures and large holes compared with uninoculated SBM by SEM.
- Chen, L.; Zhao, Z.; Yu, W.; Zheng, L.; Li, L.; Gu, W.; Xu, H.; Wei, B.; Yan, X. Nutritional quality improvement of soybean meal by Bacillus velezensis and Lactobacillus plantarum during two-stage solid-state fermentation. AMB Express. 2021, 11, 23, doi:10.1186/s13568- 021-01184-x.
- Wang, C.; Shi, C.; Su, W.; Jin, M.; Xu, B.; Hao, L.; Zhang, Y.; Lu, Z.; Wang, F.; Wang, Y., et al. Dynamics of the physico-chemical characteristics, microbiota, and metabolic functions of soybean meal and corn mixed substrates during two-stage solid-state fermentation. mSystems 2020, 5, doi:10.1128/mSystems.00501-19.
Acetic acid, which can improve the aerobic stability of fermented feed through significant antifungal properties [26], had consistently higher levels in all the treated groups than that in CK after 30 d of fermentation and subsequent aerobic exposure, which may be one of the reasons for the better aerobic stability of the treatment groups. Guan et al. [27] also found that the heterofermentative L. buchneri and L. rhamnosus improved fermented feed aerobic stability by producing higher levels of acetic acid.
- Bai, J.; Xu, D.; Xie, D.; Wang, M.; Li, Z.; Guo, X. Effects of antibacterial peptide-producing Bacillus subtilis and Lactobacillus buchneri on fermentation, aerobic stability, and microbial community of alfalfa silage. Bioresour. Technol. 2020, 315, 123881. doi:10.1016/j.biortech.2020.123881.
- Guan, H.; Ran, Q.; Li, H.; Zhang, X. Succession of microbial communities of corn silage inoculated with heterofermentative lactic acid bacteria from ensiling to aerobic exposure. Fermentation 2021, 7, 258. doi:10.3390/fermentation7040258.
and similar effects were also reported by Su et al. [28] that after SSF, the contents of CP and trichloroacetic-acid-soluble protein increased (p < 0.05) by 14.28% and 25.53%, respectively. 28. Su, W.; Jiang, Z.; Hao, L.; Li, W.; Gong, T.; Zhang, Y.; Du, S.; Wang, C.; Lu, Z.; Jin, M., et al. Variations of soybean meal and corn mixed substrates in physicochemical characteristics and microbiota during two-stage solid-state fermentation. Front. Microbiol. 2021, 12, 688839, doi:10.3389/fmicb.2021.688839.
This is also consistent with the detection of butyric acid in the CK group only. Several studies have proven that AA has a strong inhibitory effect on bacteria and fungi including Aspergillus flavus, Escherichia coli and Colletotrichum fragariae [13,31]. Thus, the ZA3+AA treatment used in the present study can inhibit undesirable bacteria and thereby reduce excessive spoilage of FSBM at the aerobic exposure stage.
- Guan, X.; Ge, D.; Li, S.; Huang, K.; Liu, J.; Li, F. Chemical composition and antimicrobial activities of Artemisia argyi Levl. et vant essential oils extracted by simultaneous distillation-extraction, subcritical extraction and hydrodistillation. Molecules 2019, 24. doi:10.3390/molecules24030483.
- Sonker, N.; Pandey, A.K.; Singh, P. Efficiency of Artemisia nilagirica (Clarke) Pamp. essential oil as a mycotoxicant against postharvest mycobiota of table grapes. J. Sci. Food Agric. 2015, 95, 1932-1939. doi:10.1002/jsfa.6901.
Point 5: I recommend do not be use abbreviation in subtitles. For example:
Line 172 – “Characteristics of SBM and AA”
Line 193 – “Sensory evaluation and surface structure of FSBM” etc.
Thank you, we have revised in whole subtitles, please see the entire revised manuscript.
Point 6: All figures should be enlarged so that the data in them can be seen.
Thanks for the good suggestion, and we have revised all figures.
Point 7: The part of Conclusion:
Just like the abstract there should be no more abbreviations in this part.
Thank you, we have revised. Please see P18, L628 to P19, L640.
